# Analyzing a networked social algorithm for collective selection of representative committees

**Alexis R. Hernández**[1]*, **Carlos Gracia-Lázaro**[2], **Edgardo Brigatti**[1], **Yamir Moreno**[2,3,4]

**1** Instituto de Física, Universidade Federal do Rio de Janeiro, Rio de Janeiro, Brazil, **2** Institute for Biocomputation and Physics of Complex Systems (BIFI), Universidad de Zaragoza, Zaragoza, Spain, **3** Department of Theoretical Physics, Faculty of Sciences, Universidad de Zaragoza, Zaragoza, Spain, **4** ISI Foundation, Turin, Italy

* elchechi@gmail.com

**Data Availability Statement:** All relevant data are within the manuscript and its Supporting Information files.

**Funding:** C. G-L and Y.M. acknowledge partial support from the Government of Aragón,

## Abstract

A recent work (Hernández, *et al.*, 2018) introduced a networked voting rule supported by a trust-based social network, where indications of possible representatives were based on individuals opinions. Individual contributions went beyond a simple vote-counting and were based on proxy voting. This mechanism selects committees with high levels of representativeness, weakening the possibility of patronage relations. By incorporating the integrity of individuals and its perception, we here address the question of the resulting committee's trustability. Our results show that this voting rule provides sufficiently small committees with high levels of representativeness and integrity. Furthermore, the voting system displays robustness to strategic and untruthful application of the voting algorithm.

## Introduction

The form of citizen participation in contemporary and complex democracies is a central issue in social debate. Many transformations and possible innovations have been recently discussed [2–4], which redesign our interactions in politics and society, often forced by the widespread use of digital technologies. A general problem, which ranges from national to neighborhood scales, is the problem of selecting an exemplary group of representatives to make decisions on behalf of the community [5–7].

Despite the prolific theoretical and philosophical debate over these issues [8, 9], examples of empirical construction of new algorithms have been relatively limited [1, 10–12]. Recently, Hernandez *et al.* introduced a new social algorithm for collective selection of a committee of representatives [1]. The algorithm is developed starting from a standard situation where each voter is allowed to vote for only one candidate. However, the elected representatives are the ones who obtain a better rank among their counterparts, in a way that individual contributions go far beyond a simple vote-counting.

The introduced formal algorithm presents new specific features which could improve governance legitimation and fairness. The lists of candidates are not fixed in advance, but they

Spain through a grant to the group FENOL (E36-17R: https://www.aragon.es/Departamentos OrganismosPublicos/Departamentos/ InnovacionInvestigacionUniversidad), and by MINECO and FEDER funds (grant FIS2017-87519-P: http://www.ciencia.gob.es/portal/site/MICINN? lang_choosen=en). Y.M. also acknowledges partial support by Intesa Sanpaolo Innovation Center. The funders had no role in study design, data collection and analysis, decision to publish, or preparation of the manuscript.

**Competing interests:** The authors have declared that no competing interests exist.

emerge as a self-organized process controlled by the voting rules. This fact introduces an effective participation and engagement of the whole community, in contrast to top-down candidate rigid lists. The voters express not preferences, but opinions, which determine their indications about whom they would like to see as their representatives. Finally, the new proposed mechanism improves the committee representativeness, weakening the possibility of patronage and clientelism relations. Additionally, the vote aggregation mechanism is supported by a self-declared confidence circle, which defines a network of trusted individuals. This trust-based social network, which can be implemented on an online platform, is a fundamental ingredient that allows for direct accountability of the elected committee. Even if based on a local network, it can naturally scale to national sizes, translating to those larger scales an effective accountability typical of small-sized communities.

In this work, we analyze a new aspect that can be introduced in the original algorithm. Specifically, we incorporate the possibility of a form of direct choice of individuals over the possible elected representatives. Hence, we mitigate the aspect that voters determine their indications about whom they would like to see as their representative through opinions, valuing the principle that individuals directly select candidates. This new ingredient is implemented by introducing a declared preference among the contact network of individuals. Preferences act as a weight on the original opinion-based ranking algorithm in such a way that higher rates for these preferences are assigned to individuals considered more apt to participate in the committee.

The described mechanism improves the legitimation, fairness, and effectiveness of the committee. In fact, overlaps, which are not controlled by voters, are weighted by a term subjectively assigned by the individuals. This weight should encourage a check on incompetence and corruption: Incompetence because an equal say for every individual is not necessarily always desired; Corruption as the preference should be proportional to the person who demonstrates and promises true integrity (sound ethical principles and trust). As each voter knows their representatives and each committee member knows whom he is accountable to, this fact allows for a strong control over representatives' actions.

The purpose of this work is to present and characterize in depth the new social algorithm throughout computational analyses. In Sec. II we describe the details of the algorithm. Sec. III is devoted to test the new voting rule, modeling the behavior of the selected committee. The quality of the elected committee is assessed looking at how much their final decisions are consistent with the community's personal opinions and estimating the general integrity of the elected committee. Finally, Sec. IV presents some discussions of our results and concluding remarks.

## The model

Let us assume a system composed by $N_e$ electors interacting on an internet-based platform. The platform allows the voters to declare who belongs to their interaction circle, rendering a network of well-known individuals. Voters also declare their perception of integrity for each individual $k$ belonging to their interaction circle. This perception is condensed in a scalar value $I_{jk} \in [0, 1]$, which represents the perception that individual $j$ has about the integrity of individual $k$.

In a following step, voters manifest their opinions on $N_i$ issues. Issues are organized in questions which can be defined by a committee or by means of a self-organized process internal to the community. The answers of each individual $j$ are organized in a vector $v^j$, composed by $N_i$ cells. Each cell assumes the value 1 for a positive answer, −1 for a negative one or 0 for a

question left unanswered. Given the previous steps, the representative of a given individual $j$ is selected by means of the following algorithm.

The vector's overlap of each individual $j$ with all his neighbors $k$ is computed through the following expression [1]:

$$v^j * v^k = \frac{\sum_{m=1}^{N_i} (v_m^j \cdot v_m^k) \delta(v_m^j, v_m^k)}{\sum_{m=1}^{N_i} (v_m^j \cdot v_m^k)^2} \ , \tag{1}$$

where the numerator counts the number of questions answered in the same way (only yes or not) and the denominator counts the number of questions answered simultaneously by both individuals; $\delta$ stands for the Kronecker delta which is 1 if $v_m^j = v_m^k$ and 0 otherwise. Then, we calculate the product of the previously defined overlap with the variable $I_{jk}$ (*i.e.*, the integrity of $k$ as perceived by $j$), obtaining the ranking function:

$$R_{jk} = I_{jk} (v^j * v^k) \tag{2}$$

The introduction of the term $I_{jk}$ establishes a form of direct choice of the individual $j$ over the possible elected representative. Overlaps, which are not controlled by voters, are weighted by a term subjectively assigned by the individuals. Notice that we are simply considering the term $I_{jk}$ associated to each agent $k$. Instead, we can consider a statistical measure over the social circle of $k$ to obtain a more precise evaluation of $k$'s integrity. However, this will introduce an external interference in $j$'s choice which can be undesirable in a democratic process. Finally, each individual $j$ will indicate as his representative the individual $k'$ for which $R_{jk'}$ is maximum. In the case where the same maximum value is shared by more than one individual, the one with a higher connectivity is selected. For the exceptional case of equal connectivity, the representative is randomly chosen between the equivalent ones. The introduction of the perception of integrity and its use in the evaluation of the ranking function is the principal novelty of this work in relation to the original algorithm of the voting rule introduced in [1].

After the selection of the representative $k'$ for every voter $j$, the final step consists of choosing the aggregate of representatives for the entire community. To this end, we construct a directed graph, which we call the representative graph, where a node represents each individual and a directed link connects the individual with his personal representative. In this graph, which in general is composed by different disconnected clusters, cycles are present. These cycles represent individuals that have been mutually indicated by themselves. Technically the representative graph is a directed graph with out-degree 1. It is composed of disconnected components, each one formed by a cycle with trees attached to the cycle nodes (see Fig 1). Considering a transitivity process, votes flow through the trees until they get to the cycles. Hence, the cycles' individuals are proper potential representatives for the community.

As a final step, among the individuals belonging to a cycle, only the ones with a number of votes larger than a threshold $\Theta$ are indicated as representatives. Votes are counted considering the cumulative flow defined by the directed graph: If the individual $j$ is pointing to $z$, $z$ receives all the votes previously received by $j$ plus one. This flow of votes is computed only following links outside the cycles. Inside the cycles, only the single vote of an individual is counted. To sum up, the votes $v$ received by an individual $i$ inside a cycle are equal to:

$$v_i = 1 + \sum_{t \in G(i)} l_t \tag{3}$$

where $G(i)$ is the set of all the trees ending at node $i$ and $l_t$ is the number of links of the tree $t$. Based on this score, the number of representatives is reduced and results in a fraction of the total number of individuals that belongs to the cycles.

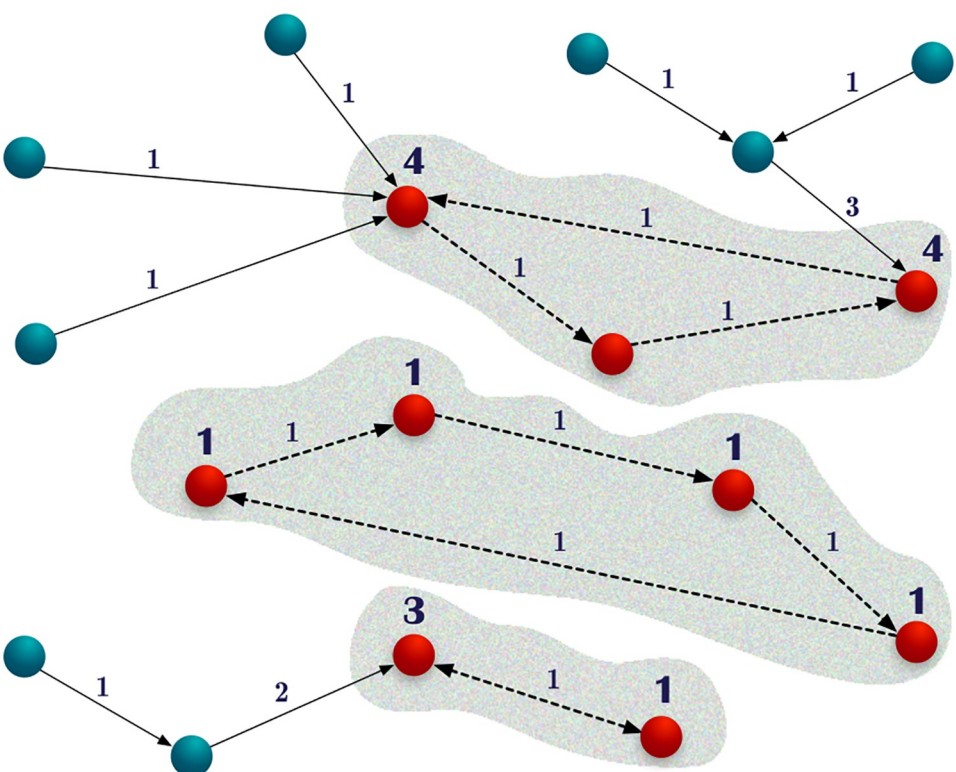

**Fig 1. Schematic representation of the vote process.** Nodes stand for the individuals; the red ones belong to a cycle and will be confirmed as representatives if they collect more votes than the established threshold. The big numbers associated to the nodes represent the received cumulated votes. Arrows stand for the indication of each individual and the small numbers associated to them represent the number of transferred votes. Dotted arrows belong to a cycle where there is no cumulative vote transfer. Figure extracted from Ref. [1].

## Results and discussion

In our simulations, each individual is assigned an intrinsic integrity $i_k$, which is a number uniformly distributed in the interval [0, 1]. The perceived integrity $I_{jk}$ corresponds to $i_k$ shifted by the error in the perception that $j$ has on the integrity of individual $k$, which is modeled by a scalar $\delta i_{j,k} \in [0, 1]$ drawn from a Gaussian distribution $N(0, \sigma_p)$. In order to keep $I_{jk} \in [0, 1]$, $I_{jk}$ values greater than 1 are set to 1 and negative values are set to 0: $I_{jk} = max[min(i_k + \delta i_{j,k}, 1), 0]$. On the other hand, the individuals' opinions in relation to the selected issues are randomly generated with the following rule: given an issue $i$, an individual does not have an opinion ($v_i = 0$) with probability 1/3. The probability to have an opinion $v_i = +1(-1)$, is $1/3 + \epsilon_i$ ($1/3 - \epsilon_i$), where $\epsilon_i$ is a random variable following a normal distribution with mean value equal to zero and $\sigma^2 = 0.05$.

The interaction circles are modeled by generating a network where nodes represent individuals and links the social relationships present in the community. The interaction circle of an individual is obtained selecting a node and considering its first neighbors. Note that an important simplification of this approach is the fact that it generates individuals with symmetric social relationships. In the following analysis three types of networks are considered. Homogeneous random networks, implementing the Erdös-Rényi model [13], where the degree distribution is peaked around a typical value $\langle k \rangle$; heterogeneous networks, using the Barabási-Albert model [14], with a power-law degree distribution $P(k) \propto k^{-3}$; and the so-called small-world Watts and Strogatz network model [15]. Our aim is not to model specific aspects

of a real social network, but to use simple examples just to discuss the possible influence of some relevant network properties on the behavior of our model (such as the heterogeneity in the degree distribution, the average degree and the small-world property).

The system can be characterized by three observables:

- The normalized committee size, which is the ratio between the number of elected individuals ($E$) and the total number of individuals of the community: $F = E/N_e$.

- The representativeness $R$, which is measured by calculating the fraction of decisions expressed by the elected committee ($e_j$) which matches with the community decisions ($c_j$) over all the considered $N_i$ issues: $R = \frac{\sum_{j=1}^{N_i} \delta(e_j - c_j)}{N_i}$. The committee's decisions are attained by means of a majority vote where each representative's vote is weighted by the number of collected votes during the election procedure. The community decision corresponds to the result of a plebiscite, where every individual votes follow the opinion expressed in his vector $v^j$ (no opinion corresponds to abstention). For $R = 1$ a perfect representativeness is obtained: a committee makes all the decisions in line with the popular will. On the opposite, for binary decisions, $R = 1/2$ corresponds to a non-representative committee, whose decisions are completely uncorrelated to the popular will. A useful observable is $1 - R$, which measures how far the system is from the perfect representativeness. This quantity is particularly interesting because, for the original model without integrity [1], it presents a simple and robust relation with $F$:

$$1 - R \propto 1/\sqrt{F} \qquad (4)$$

- The integrity $I$ which is the mean value of the intrinsic integrity $i_k$ of the individuals selected for the committee.

We perform our analysis varying the value of the threshold $\Theta$, such as to obtain committees of relatively small size but with a high representativeness level—close to 0.9—(see [1] for details). In order to explore the relation between committee size and representativeness we plot the representativeness versus the normalized committee size. As can be seen the logarithmic plot of $1 - R$ versus the normalized committee size, $F$ (Fig 2), the introduction of the integrity parameter has a marginal impact on relation 4. Only for higher values of $F$, which are unpractical, a slightly worse representativeness in relation to the classical algorithm is perceived. As for the classical algorithm, for fixed R, the normalized committee size increases with the number of issues. The integrity behavior as a function of $F$ has a quite simple response: it shows very high values and a final abrupt drop for large committee size. This is due to the probability for lower integrity individuals to obtain the necessary amount of votes to be elected becoming relevant. The dependence on $N_i$ is weak and establishes a trade-off between Representativity and Integrity. More issues make the overlap less relevant in the computation of $R_{ik}$ improving the integrity at the expenses of the representativity.

The dependence of the above observables with the system size $N_e$ (Fig 3) shows that the latter has an impact on the representativeness but not on the integrity behavior. In fact, as it was the case in the original model, when fixing R the committee size decreases for larger system sizes. For example, for the parameters used in Fig 3, a representativity of 0.9 corresponds to a committee of 78 members for a community of 2500 individuals, and to 36 representatives for $N_e = 40000$. Furthermore, as can be seen in Fig 4, the error in integrity perception, which is controlled by the parameter $\sigma_p$, has no effect on the representativeness. In contrast, it obviously affects the committees' integrity. The plateaux values of $I$ decrease with $\sigma_p$, following a simple

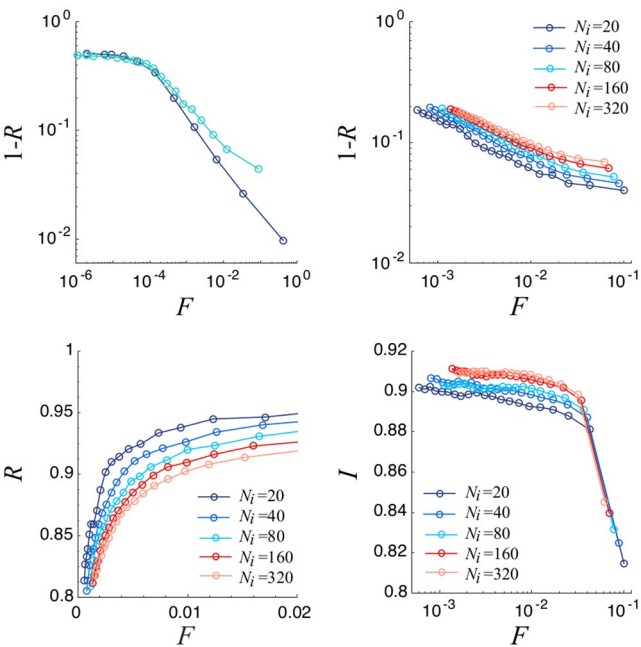

**Fig 2.** Top: On the left, logarithmic plot of $1 - R$ versus normalized committee size for the $NVR$ proposed in [1] (dark blue) and the one proposed here ($NVR_I$, light blue) with $Ni = 40$. On the right, $1 - R$ versus normalized committee size for different $N_i$ values. Bottom: Representativity (left) and Mean Committee Integrity (right) as a function of normalized committee size. We consider a Erdös-Rényi network with $N_e = 10000$, $\langle k \rangle = 40$ and $\sigma^2 = \sigma_p^2 = 0.05$. Results are averaged over 100 different realizations.

linear dependence on this parameter. Higher values of errors in the integrity perception correspond linearly to worse values in the integrity selection (see inset in Fig 4).

In Fig 5, we can see that the representativeness is not strongly dependent on network connectivity. For sufficiently high $\langle k \rangle$, the curves show the same behavior. The heterogeneity in the degree distribution of the network marginally impacts the results. For high values of $F$, the

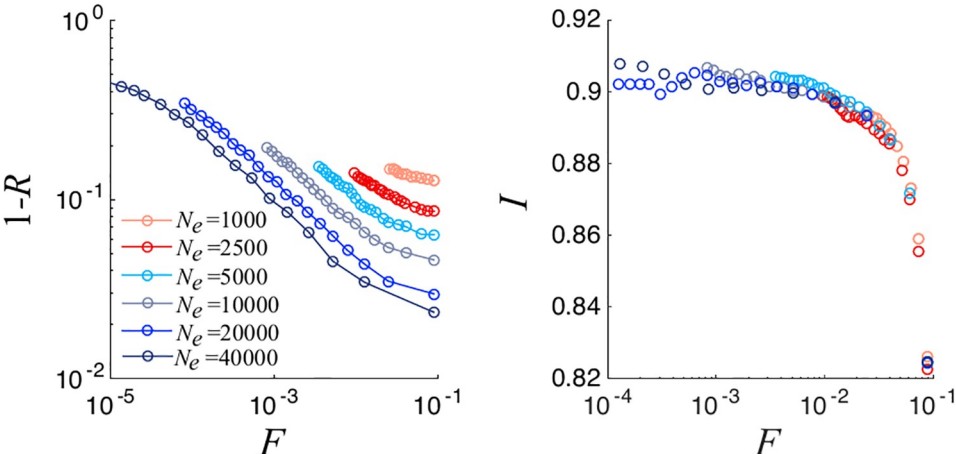

**Fig 3.** Logarithmic plot of $1 - R$ versus normalized committee size (left). Mean Committee integrity as a function of normalized committee size (right), for different numbers of electors $N_e$. We consider a Erdös-Rényi network with $N_e = 10000$, $N_i = 40$ and $\sigma^2 = \sigma_p^2 = 0.05$. Results are averaged over 100 different realizations.

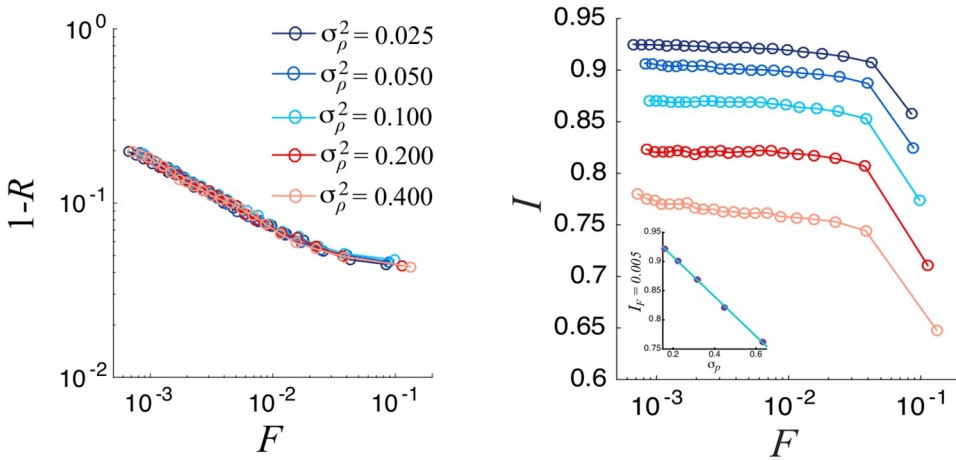

**Fig 4.** Logarithmic plot of $1 - R$ versus normalized committee size (left). Mean Committee Integrity as a function of normalized committee size (right). In the inset we show the linear behavior of $I_{F=0.005}$ with $\sigma_p$ ($I_{F=0.005} = -0.34\sigma_p + 0.98$). We consider a Erdös-Rényi network with $N_e = 10000$, $N_i = 40$, $\sigma^2 = 0.05$ and $\langle k \rangle = 40$. Results are averaged over 100 different realizations.

Barabási-Albert network performs moderately worse than Erdös-Rényi's. As in the case of the original algorithm, higher connectivity generates a small bias in the selection of the more representative individuals. In contrast, the small world property of the Watts-Strogatz network positively influences the algorithm, allowing for slightly better results in terms of representativeness. This last behavior is more pronounced than in the case of the original algorithm. We

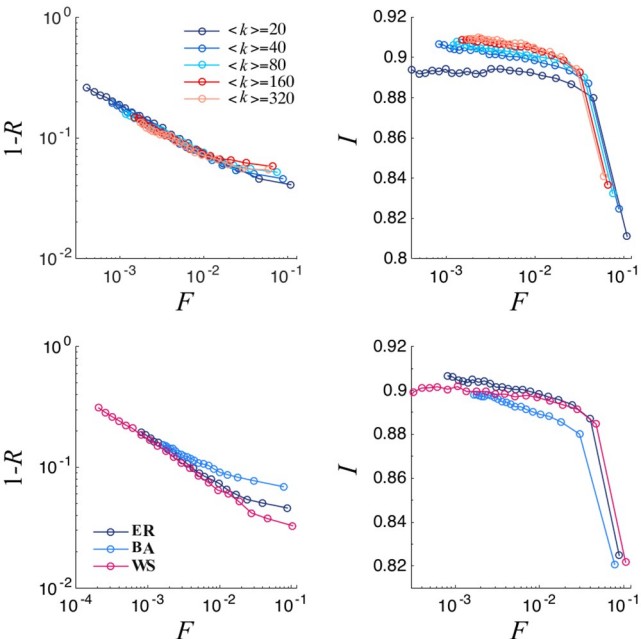

**Fig 5.** Top: Logarithmic plot of $1 - R$ versus normalized committee size (left) and Mean Committee Integrity as a function of normalized committee size (right) for different values of $\langle k \rangle$. Bottom: Logarithmic plot of $1 - R$ versus normalized committee size (left) and Mean Committee Integrity as a function of normalized committee size (right) for different network topologies. We consider a Erdös-Rényi network with $N_i = 40$, $\langle k \rangle = 40$ and $\sigma^2 = \sigma_p^2 = 0.05$. Results are averaged over 100 different realizations.

have also analyzed the impact that the presence of community structure can have on the outcome of the committee selection algorithm. We implemented the stochastic block model [16], varying the intra- and inter-community link densities and the number of communities, and we were not able to evidence relevant trends on the general results of our algorithm (see the Supplementary Material for more details).

In addition to the so far discussed synthetic networks, we have tested our voting rule via some data-driven simulations where real social networks are taken as an underlying structure and the committee selection process is implemented on the top of these real networks. We used collected data from the music streaming service Deezer, collected at November 2017 [17, 18]. This dataset represents the friendships networks of users from three European countries. Nodes represent the users and edges are the mutual friendships. We used the data relative to Croatia. They contain 54573 users and 498202 friendship relations, which correspond to $\langle k \rangle \approx 9$, a reasonable mean connectivity. We also consider data from the free on-line social network Orkut. Orkut allows users to form groups which external members can join. We used data collected by Alan Mislove *et al.* [17, 19]; they contain 3072626 nodes and 11718083 edges ($\langle k \rangle \approx 38$). In this two scenarios, we interpret the friendship relations as a social tie, which renders a network of well-known individuals based on these real data. The integrity and the opinion vector of each individual are synthetically generated following the previously implemented rules. Finally we simulated the voting process on these real social networks, in which the community elects a committee. Results, displayed in Fig 6, appear to be similar to the ones obtained with synthetic networks.

In order to compare our model's behavior with other traditional methods of representatives selection, we analyze the representativeness and the integrity of equally sized committees. A widespread method is a traditional majority voting rule (TMV) for electing representatives in a closed list of previously determined candidates. In our implementation, a list of $Nc$

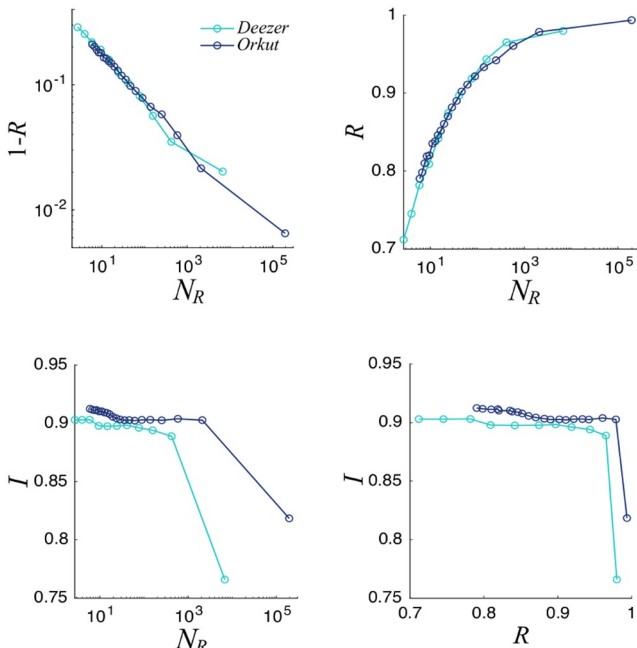

**Fig 6.** Top: On the left, logarithmic plot of $1 - R$ versus committee size. On the right, $R$ versus committee size. Bottom: Commitee Integrity as a function of $R$ (left) and committee size (right). We consider the two social networks, Deezer and Orkut. Results are averaged over 100 different realizations; $\sigma^2 = \sigma_p^2 = 0.05$.

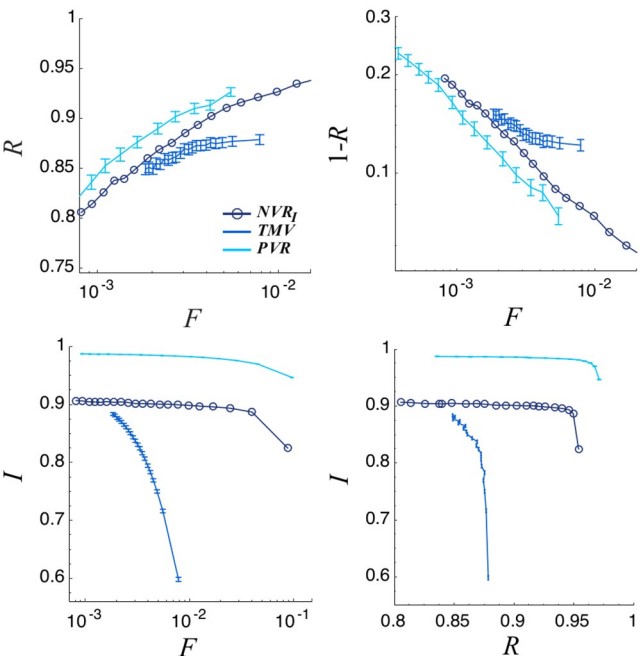

**Fig 7.** Top: Representativity versus normalized committee size (left), logarithmic plot of $1 - R$ versus normalized committee size (right). Bottom: Mean Committee integrity as a function of normalized committee size (left) and Representativity (right). The results correspond to the Networked Voting Rule ($NVR_I$), the Traditional Majority Voting ($TMV$) and a Perfect Voting Rule ($PVR$) with parameters $N_e = 10000$, $N_i = 40$ and $\sigma^2 = \sigma_p^2 = 0.05$. For the $NVR_I$ we use a Erdös-Rényi network with $\langle k \rangle = 40$. For the $TVR$ we set the initial number of candidates $N_c = 100$. Results are averaged over 100 different realizations. See the main text for a detailed explanation of the voting rules.

candidates in the community is randomly selected and each individual $j$ votes for the candidate who presents the higher $R_{jk^*}$ value ($k^*$ belongs to the list of $Nc$ candidates). Note that in this case the integrity evaluation is influenced by errors in perception. Decisions are taken with the same weighted voting rule. This modeling approach mimics a voter who has a perfect knowledge of the candidates, assuming he makes a rational decision to maximize his representation. For this voting rule, representativeness is also computed by comparing the decisions taken by the committee, obtained with a weighted majority voting process, with the results of direct popular vote. As can be appreciated in Fig 7, our model is by far more efficient, halving the committees' size and showing a better selection of representatives integrity.

Finally, we compare our method to an idealized perfect voting rule (PVR). This rule represents a situation of rational individuals that have a perfect knowledge of all others and their opinions. Moreover, they are globally networked, having direct access to all others and allowing their acts to be checked. In this situation, a voter indicates the individual with the highest overlap with his opinion vector and the best integrity (the higher $R_{jk^*}$ value). Note that in this case the evaluation of the integrity is not influenced by errors in perception. The selected committee is formed by the first $F \cdot N_e$ individuals which poll more. In this case, the committee decisions are also taken as a weighted majority vote. This voting rule, although unrealistic, is useful in at least two respects: First, very small communities can exhibit similar characteristics; Second, the model is a useful yardstick for evaluating the levels of representativeness of more realistic models. The relation between representativeness and committee sizes can be compared also in the case of the PVR rule (see Fig 7). It is quite impressive that the representativeness of our networked voting rule is comparable with the perfect one. The PVR rule is able to select a committee with a higher integrity score, but this is only possible because in this

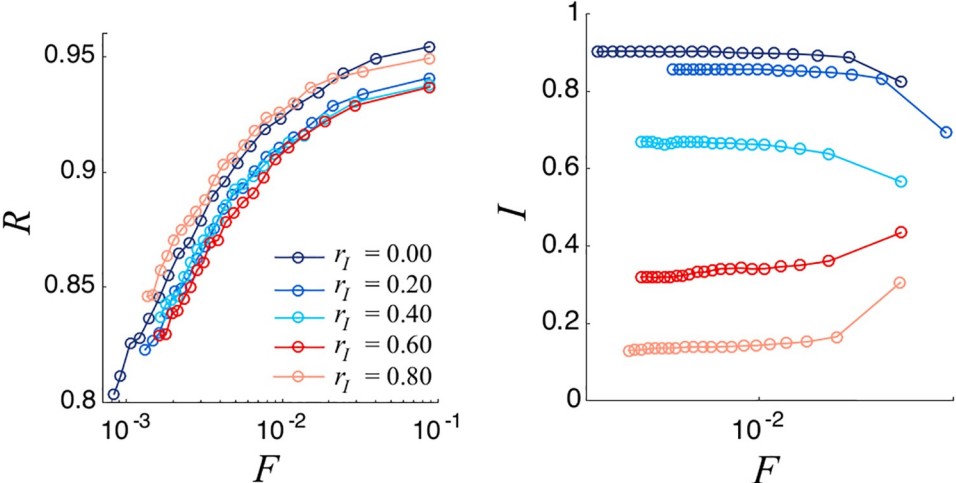

**Fig 8. Representativity versus normalized committee size (left), Mean Committee integrity as a function of normalized committee size (right) for different values of $p$ (the percentage of individuals with a distorted integrity perception).** We consider a Erdös-Rényi network with $N_e = 10000$, $\langle k \rangle = 40$ and $\sigma^2 = \sigma_p^2 = 0.05$. Results are averaged over 100 different realizations.

situation the integrity of every member is tested, and not only the integrity of a small subset, as it happens for our networked rule.

We conclude our analysis testing the resilience of our networked voting rule to possible attacks. We consider the situation in which a group of voters decides to assign high $I_{jk}$ scores to some individuals who, in contrast, are characterized by a low personal integrity. This behavior can model patronage and clientelism relations, where individuals with low integrity organize a network of social relationships for obtaining political support. In our model this behavior can be modeled introducing a percentage of individuals $p$ for whom $I_{jk} = 1 - I_{jk}$. As can be seen in Fig 8, representativeness is not seriously affected by this action. In contrast, the integrity of the elected committee is strongly influenced by this ill behavior. By fixing the normalized committee size $F$, the integrity undergoes an abrupt transition from high values towards very low values as $p$ increases, see Fig 9. Finally, we analyzed if the presence of individuals who refuse to join the elected committee can impact our results. Even if 80% of the population do not accept to be elected, results are substantially unaltered. Similarly, allowing individuals to vote for themselves does not have a significant impact on our voting rule (see the Supplementary Material for more details on these points).

## Conclusion

We analyzed a new voting algorithm, particularly well suited for online social networks, for selecting a committee of representatives with the aim of enhancing the participation of a community both as electors and as representatives. This voting system is based on the idea of transferring votes through a path over the social network (proxy-voting systems). Votes are determined by an algorithm which weights the similarity of individuals opinions and the trust between individuals directly connected in a specific social network.

Our computational analyses suggests that this voting algorithm can generate high representativeness for relatively small committees characterized by a high level of integrity. Results of representativeness and integrity are comparable with a theoretically defined perfect voting rule and, in general, perform better than a traditional voting rule with a closed

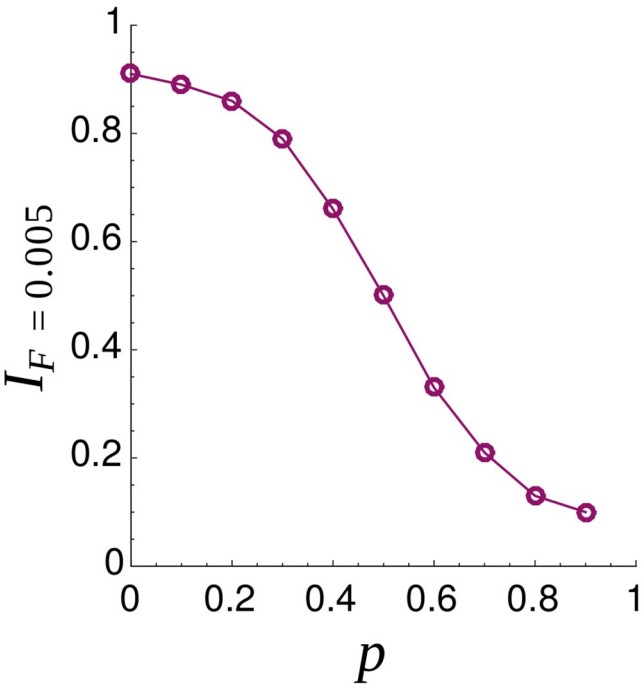

**Fig 9. Mean Committee Integrity for a normalized committee size $F = 0.005$ ($I_{F=0.005}$) as a function of the percentage of individuals with a distorted perception of the integrity ($p$).** We consider a Erdös-Rényi network with $N_e = 10000$, $\langle k \rangle = 40$ and $\sigma^2 = \sigma_p^2 = 0.05$. Results are averaged over 100 different realizations.

list of candidates. The introduction of a term which expresses the trust on the candidate's integrity does not significantly impact the representativeness of the committee, in particular for committees of small and medium sizes. The rule shows a robust dependence on community size. Besides, the perception of individual integrity directly influences the committee's quality: higher error values in the integrity perception linearly correspond to poorer values in the committees' integrity. On the other hand, representativeness is not strongly influenced by integrity perception.

Interestingly enough, these findings are not strongly dependent on the general properties of the network used to describe the community of voters, as shown by the analysis of networks characterized by different topologies. Finally, the voting system seems robust to strategic and untruthful application of the voting algorithm. In fact, even with a 20% of the votes produced by individuals which vote for candidates with a low personal integrity, the integrity of the committee is substantially unaltered, and only if unfair votes are around 40% an abrupt change is observed. In conclusion, we believe that the proposed voting rule, which fixes a particular way for the voters to express their preferences and defines a clear algorithm for determining the final identification of the committee, could be implemented in practice. If our results are confirmed in such hypothetical scenario, the algorithm discussed here will define an efficient form of democracy by delegation based on proxy voting [12], which robustly shows a high level of representativeness and integrity of the selected committee. These results are important improvements over the original voting rule introduced in [1], as by incorporating the integrity of individuals and its perception, we can address the important problem of the committee's trustability without compromising the high level of representativeness already shown by the original algorithm.

## Supporting information

**S1 Text. Supplementary Material: Analyzing a networked social algorithm for collective selection of representative committees.**
(PDF)

## Acknowledgments

We would like to thank Gabriel M. Lando for a critical reading which improved the readability of our manuscript.

## Author Contributions

**Conceptualization:** Alexis R. Hernández, Carlos Gracia-Lázaro, Edgardo Brigatti, Yamir Moreno.

**Formal analysis:** Alexis R. Hernández, Carlos Gracia-Lázaro, Edgardo Brigatti, Yamir Moreno.

**Writing – original draft:** Alexis R. Hernández, Edgardo Brigatti.

**Writing – review & editing:** Alexis R. Hernández, Carlos Gracia-Lázaro, Edgardo Brigatti, Yamir Moreno.

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
