## [Decision Letter · Decision Letter 0]

18 Jul 2019

PONE-D-19-14447

Analyzing a networked social algorithm for collective selection of representative committees

PLOS ONE

Dear Dr. Hernández,

Thank you for submitting your manuscript to PLOS ONE. After careful consideration, we feel that it has merit but does not fully meet PLOS ONE’s publication criteria as it currently stands. Therefore, we invite you to submit a revised version of the manuscript that addresses the points raised during the review process.

ACADEMIC EDITOR: Please insert comments here and delete this placeholder text when finished. Be sure to:

Indicate which changes are required versus recommended for acceptanceAddress any conflicts between the reviewsProvide specific feedback from your evaluation of the manuscript

We would appreciate receiving your revised manuscript by Aug 25 2019 11:59PM. To enhance the reproducibility of your results, we recommend that if applicable you deposit your laboratory protocols in protocols.io, where a protocol can be assigned its own identifier (DOI) such that it can be cited independently in the future. For instructions see: http://journals.plos.org/plosone/s/submission-guidelines#loc-laboratory-protocols

We look forward to receiving your revised manuscript.

Kind regards,

Ginestra Bianconi

Academic Editor

PLOS ONE

Journal Requirements:

Additional Editor Comments (if provided):

Reviewers' comments:

Reviewer's Responses to Questions

**Comments to the Author**

1. Is the manuscript technically sound, and do the data support the conclusions?

Reviewer #1: Yes

2. Has the statistical analysis been performed appropriately and rigorously? 

Reviewer #1: Yes

3. Have the authors made all data underlying the findings in their manuscript fully available?

Reviewer #1: Yes

4. Is the manuscript presented in an intelligible fashion and written in standard English?

Reviewer #1: Yes

5. Review Comments to the Author

Reviewer #1: I have read carefully the paper entitled as “Analyzing a networked social algorithm for collective selection of representative committees“ by Hernández et al., submitted to PLoS ONE for publication. The paper builds on an earlier work published by the same set of authors recently [1], which proposes an algorithm to construct representative committees using personal and collective preferences in networked populations. In this paper they extend their earlier findings by considering committee’s trustability.

I have some comments what I suggest to the authors to consider:

- Figure 1 has been already published in [1] and this fact is not acknowledged in the actual manuscript.

- It is not clear what is novel algorithmically and in terms of results as compared to the pervious paper of the authors [1]. Please highlight.

- Their method assumes global network knowledge about the underlying social network what is commonly unknown. The authors (implicitly) argue that this is not a problem as their method is meant for online social networks where social ties are mapped with high precision. However, it is usually not the case as (a) detailed online social network data is not available but only for the provider, (b) it may contain several non-real social ties and non-human actors (e.g. bots), and (c) it may not capture all social ties (e.g. offline relationships) which at the same time might be important for opinion formation. I would suggest to the authors to address these questions and show how the outcome of their process is changing by assuming incomplete knowledge about the network structure.

- In page 4 the authors explain that a representative committee can be selected in two steps: first identifying cycles in the representative graphs and then by thresholding to select people by the number of votes they gained in their downstream tree structure.

In my opinion, it is a possible scenario that a group of people agree in advance to bias the first step of this process to vote such that they form a cycle. This way they would increase the probability that some of them will be selected from the cycle in the committee. The authors address resilience issues in the end of the manuscript (starting from line 218) but miss to address the problem when fraud is not individual but organised between a larger group of people.

- In the paragraph starting from line 119 the authors discuss that they tested their algorithm on three conventional network models while they were concentrating on the dependencies of the selection outcome on generic network properties like degree heterogeneity, average connectivity, or shortest paths. One important characteristic missed here is community structure, which can largely influence the outcome of the committee selection algorithm. I would suggest to the authors to use one of the many (Planted L-partition model, NG benchmark, LFR benchmark) community network model to test the effect of intra/inter community link density on the outcomes.

- For validation purposes it would be necessary that the authors explore their model via data-driven simulations where they take a real social network as an underlying structure and model the committee selection process on the top. Simulating the process only on synthetic overly simplified network models is important for exploration but may provide results far from reality.

Typos:

Erdos -> Erd\\H{o}s

Barabasi -> Barab\\'asi

6. PLOS authors have the option to publish the peer review history of their article (what does this mean?). If published, this will include your full peer review and any attached files.

Reviewer #1: No

---

## [Author Response · Author response to Decision Letter 0]

29 Aug 2019

Dear Editor,

Please, find enclosed a revised version for our manuscript "Analyzing a networked social algorithm for collective selection of representative committees", co-authored by Alexis R. Hernández, Carlos Gracia-Lazaro, Edgardo Brigatti and Yamir Moreno, which we are resubmitting for publication in Plos One.

We would like to thank the reviewer for his/her favorable opinion on our work as well as for providing very useful feedback that has led to an improved version of our contribution. Aside from the necessary modifications to the manuscript, which we detail below, we are enclosing here a detailed response to the reviewer's comments. We have gone through all his/her points and suggestions and we believe he/she will be satisfied with our answers and with the corresponding improvements to the manuscript.

We also provide below a list of changes, summarizing the modifications to the manuscript, which are described in more detail in the answers to the reviewer. 

This manuscript describes original work and is not under consideration by any other journal. All authors approved the new version of the manuscript and this submission.

Yours sincerely,

Alexis R Hernández, on behalf of all authors.

Instituto de Física, Universidade Federal do Rio de Janeiro,

Rio de Janeiro, Brazil.

List of Changes (summary):

 • In line 85, we clarify the algorithmic contribution of the paper.

 • In Fig. 1 caption we indicate that figure 1 was extracted from Ref. 1.

 • In line 189 we mention the results obtained for modular networks.

 • In line 196 we describe the results considering real networks. These results are reported in the new Fig. 6.

 • In line 253 we slightly modified the text and figures citation.

 • In line 258 we point out the results related to unavailable representatives. 

 • In line 261 we point out the results related to self-declared candidates. 

 • In line 295 we clarify the main contribution of our work.

 • In line 301 we introduce the supplementary information section.

 • We add a reference on block models (Ref. 16).

 • Suggested and other typos were corrected. 

Answer to Reviewer #1 

We thank very much the reviewer for his/her very positive assessment of our manuscript and the feedback provided. Please, find below a detailed response to all comments received as well as the changes we have done to implement your suggestions whenever needed. 

Reviewer #1: 

I have read carefully the paper entitled as “Analyzing a networked social algorithm for collective selection of representative committees“ by Hernández et al., submitted to PLoS ONE for publication. The paper builds on an earlier work published by the same set of authors recently [1], which proposes an algorithm to construct representative committees using personal and collective preferences in networked populations. In this paper they extend their earlier findings by considering committee’s trustability.

I have some comments what I suggest to the authors to consider:

- Figure 1 has been already published in [1] and this fact is not acknowledged in the actual manuscript.

R: Thanks for pointing out this issue, we corrected that in the revised version.

- It is not clear what is novel algorithmically and in terms of results as compared to the previous paper of the authors [1]. Please highlight.

R: The new ingredient in the algorithm appears in equation (2) where the overlap is multiplied by the perceived integrity. In the previous paper, the transfer of votes depends only on opinion’s overlap. The introduction of this term allows us to study the performance of the algorithm with respect to the committee’s integrity. We write it explicitly on the revised version in order to make this clearer.

- Their method assumes global network knowledge about the underlying social network what is commonly unknown. The authors (implicitly) argue that this is not a problem as their method is meant for online social networks where social ties are mapped with high precision. However, it is usually not the case as (a) detailed online social network data is not available but only for the provider, (b) it may contain several non-real social ties and non-human actors (e.g. bots), and (c) it may not capture all social ties (e.g. offline relationships) which at the same time might be important for opinion formation. I would suggest to the authors to address these questions and show how the outcome of their process is changing by assuming incomplete knowledge about the network structure.

R: Thanks for pointing out this very important issue. In order to illustrate the robustness of the algorithm, we ran new simulations where we considered a fraction of users as unavailable (in the sense of participating as representatives). In practice, it washes out the participation of a fraction of the committee representatives with their respective voting trees. Our results show that even for very high values of this fraction (0.8) the algorithm performs pretty well. These results makes it possible to reasonably think that incomplete knowledge of the network structure would not jeopardize our conclusions. We mention these results in the body of our manuscript and include figures and description in the new supplementary material.

- In page 4 the authors explain that a representative committee can be selected in two steps: first identifying cycles in the representative graphs and then by thresholding to select people by the number of votes they gained in their downstream tree structure.

In my opinion, it is a possible scenario that a group of people agree in advance to bias the first step of this process to vote such that they form a cycle. This way they would increase the probability that some of them will be selected from the cycle in the committee. The authors address resilience issues in the end of the manuscript (starting from line 218) but miss to address the problem when fraud is not individual but organized between a larger group of people.

R: Thanks for raising this interesting question. To address this issue we considered a slightly different situation. We introduced a fraction of individuals who refuse to transfer their votes, making them self-candidates to the resulting committee. Again we observe the system to be very robust and even for a very high fraction of self-declared individuals (0.8), we find it to behave properly, achieving committees with high levels of representativity and integrity. These results are also included in the new supplementary material and commented on the body of our manuscript.

- In the paragraph starting from line 119 the authors discuss that they tested their algorithm on three conventional network models while they were concentrating on the dependencies of the selection outcome on generic network properties like degree heterogeneity, average connectivity, or shortest paths. One important characteristic missed here is community structure, which can largely influence the outcome of the committee selection algorithm. I would suggest to the authors to use one of the many (Planted L-partition model, NG benchmark, LFR benchmark) community network model to test the effect of intra/inter community link density on the outcomes.

R: Thanks again, this is also a very interesting point. In order to answer this question, we generated random networks with a specific number of communities and modularity. The results are described in the new supplementary material and commented on the body of the article. Again the system presents a robust behavior allowing the conformation of a committee with high values of representativity and integrity even for highly modular networks. 

- For validation purposes it would be necessary that the authors explore their model via data-driven simulations where they take a real social network as an underlying structure and model the committee selection process on the top. Simulating the process only on synthetic overly simplified network models is important for exploration but may provide results far from reality.

R: Thanks again. We ran new simulations considering the network structure of two online social networks: one from a music streaming service (Deezer) and the other from a free online social network (Orkut). In both cases the results are consistent with the ones obtained for synthetic networks. We include these results in the manuscript.

Typos:

Erdos -> Erd\\H{o}s

Barabasi -> Barab\\'asi

R: Typos where corrected, thank you very much for pointing out.

---

## [Editor Report · Decision Letter 1]

11 Sep 2019

Analyzing a networked social algorithm for collective selection of representative committees

PONE-D-19-14447R1

Dear Dr. Hernández,

We are pleased to inform you that your manuscript has been judged scientifically suitable for publication and will be formally accepted for publication once it complies with all outstanding technical requirements.

With kind regards,

Ginestra Bianconi

Academic Editor

PLOS ONE
---

## [Editor Report · Acceptance letter]

13 Sep 2019

PCOMPBIOL-D-19-00976R2 

Analyzing a networked social algorithm for collective selection of representative committees 

Dear Dr. Hernández:

I am pleased to inform you that your manuscript has been deemed suitable for publication in PLOS ONE. Congratulations! Your manuscript is now with our production department. 

With kind regards,

on behalf of

Dr. Ginestra Bianconi 

Academic Editor

PLOS ONE